# Course of recovery of respiratory muscle strength and its associations with exercise capacity and handgrip strength: A prospective cohort study among survivors of critical illness

**Mel Major** [1,2,3,4]*, **Maarten van Egmond**[1,2,4], **Daniela Dettling-Ihnenfeldt**[3], **Stephan Ramaekers**[2,3], **Raoul Engelbert**[2,3], **Marike van der Schaaf**[2,3,4]

1 European School of Physiotherapy, Faculty of Health, Amsterdam University of Applied Sciences, Amsterdam, The Netherlands, 2 Center of Expertise Urban Vitality, Faculty of Health, Amsterdam University of Applied Sciences, Amsterdam, The Netherlands, 3 Amsterdam UMC, location University of Amsterdam, Rehabilitation Medicine, Amsterdam, The Netherlands, 4 Amsterdam Movement Sciences, Ageing and Vitality, Amsterdam, The Netherlands

* m.major@hva.nl

## Abstract

### Background

Mechanical ventilation affects the respiratory muscles, but little is known about long-term recovery of respiratory muscle weakness (RMW) and potential associations with physical functioning in survivors of critical illness. The aim of this study was to investigate the course of recovery of RMW and its association with functional outcomes in patients who received mechanical ventilation.

### Methods

We conducted a prospective cohort study with 6-month follow-up among survivors of critical illness who received $\geq$ 48 hours of invasive mechanical ventilation. Primary outcomes, measured at 3 timepoints, were maximal inspiratory and expiratory pressures (MIP/MEP). Secondary outcomes were functional exercise capacity (FEC) and handgrip strength (HGS). Longitudinal changes in outcomes and potential associations between MIP/MEP, predictor variables, and secondary outcomes were investigated through linear mixed model analysis.

### Results

A total of 59 participants (male: 64%, median age [IQR]: 62 [53–66]) were included in this study with a median (IQR) ICU and hospital length of stay of 11 (8–21) and 35 (21–52) days respectively. While all measures were well below predicted values at hospital discharge (MIP: 68.4%, MEP 76.0%, HGS 73.3% of predicted and FEC 54.8 steps/2m), significant 6-month recovery was seen for all outcomes. Multivariate analyses showed longitudinal associations between older age and decreased MIP and FEC, and longer hospital length of stay and decreased MIP and HGS outcomes. In crude models, significant, longitudinal

**Data Availability Statement:** Data will be held in the public repository Figshare (DOI: 10.21943/auas.20798332).

**Funding:** MM received a research grant by the Dutch Research Council (NWO, 023.007.006). The funder had no role in study design, data collection and analysis, decision to publish or preparation of this manuscript. Part of this research project was funded by Taskforce for Applied Research (SIA, RAAK.PUB04.037). This funder also did not have a role in study design, data collection and analysis, decision to publish, or preparation of the manuscript.

**Competing interests:** The authors have declared that they have no competing interests

associations were found between MIP/MEP and FEC and HGS outcomes. While these associations remained in most adjusted models, an interaction effect was observed for sex.

## Conclusion

RMW was observed directly after hospital discharge while 6-month recovery to predicted values was noted for all outcomes. Longitudinal associations were found between MIP and MEP and more commonly used measures for physical functioning, highlighting the need for continued assessment of respiratory muscle strength in deconditioned patients who are discharged from ICU. The potential of targeted training extending beyond ICU and hospital discharge should be further explored.

## Introduction

Critical illness and medical treatments in the intensive care unit (ICU) impact on physiological and psychological functioning. Due to medical and technological advancements, interventions in the ICU are often lifesaving. Recovery of critical illness is, nevertheless, challenging and often incomplete [1–3]. ICU-acquired weakness (ICU-AW) is one of the major physical consequences resulting from the combination of critical illness, sedation, mechanical ventilation, and immobilization [4–7]. Several underlying mechanisms, including immobility and catabolic processes, lead to mitochondrial loss and dysfunction which cause a decrease in muscle mass and impaired contractile muscle function [8, 9]. This catabolic state might have started prior to critical illness and ICU admission and may extend beyond ICU discharge [10]. Most patients who are mechanically ventilated develop ICU-acquired *respiratory* weakness, which can contribute to failed weaning attempts, prolonged ICU-stay, and reduced chances of survival [11, 12]. Respiratory muscle weakness (RMW) is distinguished into dysfunction of inspiratory and expiratory muscles, which can be evaluated with measurements of maximal static inspiratory pressure (MIP, or PImax) and maximal static expiratory pressure (MEP, or PEmax) [13, 14].

The diaphragm, scalene and external intercostal muscles are primarily responsible for generating the inspiratory force while the abdominal wall muscles and internal intercostal muscles generate most of the expiratory force [11, 15]. Prevalence of ICU-diaphragm dysfunction (ICU-DD) can be as high as 80%, initiates after the start of mechanical ventilation and is associated with poor outcome [11, 12, 16]. The long-term prognosis of patients with concurrent presentation of ICU-AW and ICU-DD is worse compared to patients with independent ICU-AW or ICU-DD [17, 18].

While risk factors for RMW and its relationship with poor outcome are increasingly recognized, limited investigation has been conducted on prevention, treatment, and recovery over time [11, 12, 15, 19]. Most studies on RMW in critically ill patients are conducted during spontaneous breathing trials or directly after extubation, but data on prevalence of RMW after ICU and hospital discharge is lacking. A recent study reported that RMW at time of ICU discharge is associated with a decrease in handgrip strength (HGS), physical functioning, and quality of life (QoL) up to 5 years after the ICU stay [19], confirming the need for early identification of RMW in survivors of critical illness [11]. So far, longitudinal studies on the course of recovery of MIP and MEP in survivors of critical illness have not been reported [15, 19] and although interventions targeting RMW have become increasingly common within the ICU [20–22], they seldom continue after ICU and hospital discharge [23]. If longitudinal data existed on

MIP and MEP and potential associations with other functional outcomes in survivors of critical illness, we could determine if, and when, tailored respiratory muscle training could be valuable to include in post-ICU rehabilitation interventions, as is recommended by recent Delphi studies [24, 25].

Therefore, this study investigated the course of recovery of respiratory muscle strength and its associations with functional outcomes up to 6 months follow-up, in patients who received mechanical ventilation in the ICU.

## Materials and methods

A prospective cohort study with a 6-month follow up was conducted among patients who had received invasive mechanical ventilation in the ICU and were discharged from hospital. The study was performed between April 2019 and February 2021. Baseline parameters were obtained within one week after hospital discharge, with follow-up data collected at 3 and 6 months.

### Setting

We recruited participants from 2 university and 5 general hospitals in the area surrounding Amsterdam, the Netherlands. Recruitment ran concurrently with a pilot feasibility study of the department of rehabilitation medicine at the Amsterdam University Medical Centers (AMC) [26].

### Participants

Participants who received invasive mechanical ventilation (MV) for $\geq$ 48 hours in the ICUs of one of the participating hospitals and were discharged with a referral for follow-up physical therapy (PT) were eligible for inclusion. The need for follow-up PT was determined by the presence of ICU-AW (Medical Research Council Sum Score < 48), decreased physical function (Functional Ambulation Categories $\leq$ 4), dependency in Activities of Daily Living (ADLs) and/or general deconditioning. Exclusion criteria were presence of serious (preexisting) cognitive and/or psychiatric impairments or insufficient Dutch or English language skills. Potential participants were contacted within 48 hours after hospital discharge by the primary investigator (MM). After further information on the aim of the study was provided, eligibility was confirmed, and baseline measurements were scheduled.

### Data collection

The site of data collection depended on the participants' location during scheduled measurements and occurred either at the participants' homes or at a rehabilitation facility. Baseline (T0) and follow-up visits (T1 and T2) took place within one week, 3 months and 6 months after hospital discharge, respectively.

At T0, participants provided data on the following characteristics: age, sex, educational level, admission diagnosis, ICU- and hospital length of stay (LOS), duration of mechanical ventilation (MV), discharge location and follow-up referral, current medication use, and nutritional status. Data related to the ICU admission were cross-checked with the ICU-PT of the discharging hospital in situations where participants or their relatives were unsure.

Primary outcome variables were maximum inspiratory pressure (MIP) and maximum expiratory pressure (MEP). Secondary outcome variables were functional exercise capacity (FEC) and handgrip strength (HGS).

The following potential predictor variables were identified a priori: ICU LOS, MV days, hospital LOS, and age.

## Measurements

Respiratory muscle function was measured using tests of MIP and MEP with the microRPM spirometer (Micro Medical, Yorba Linda, CA, USA), which has shown to have excellent reliability (ICC > 0.90) [27]. These tests can be used as a screening tool to detect respiratory muscle weakness, show good within-subject responsiveness and reference values are available [13, 14, 28]. Three to five maneuvers were completed, until the difference between the two highest maneuvers was ≤ 10%. The highest MIP and MEP values (expressed as cmH2O) were recorded and converted into individual percentages of predicted values (% predicted) corrected for age and sex [28].

FEC was tested with the two-minute step test (TMST). The TMST is developed as part of the Senior Fitness Test [29], validated against other (functional) exercise capacity tests, reliable (ICC 0.90) in older adults with and without morbidities and practical to use in the home environment. The test can be safely conducted in patients using walking aids [29, 30] or when other functional walking tests are not physically possible. Prior to the test, safety is established by testing vital parameters such as blood pressure (< 180/100), resting heart rate (< 110 beats per minute), oxygen saturation (< 90%) and/or the presence of chronic heart failure, chest pain or dizziness. Individuals are instructed to march in place, raising the knee to a set criterion height: the midpoint between the upper margin of the patella and the anterior superior iliac spine of the ilium, which is measured for each individual and marked on the wall or a table. The goal is to complete as many steps as possible in two minutes, without running. Outcomes are expressed as the 2-minute total number of steps of the right leg reaching criterion height [29]. As reference values exist only for the population aged 60 to 94 years old, outcome data are reported as the total 2-minute step counts.

HGS was measured with the Jamar hydraulic hand dynamometer (Lafayette Instrument Company, Lafayette, IN, USA), expressed in kilograms. HGS is a commonly measured outcome in observational studies on survivors of critical illness and indicative of the presence of ICU-AW [31]. Three trials were performed bilaterally, the highest value of the dominant hand was converted to individual percentages of predicted values, corrected for age and sex [32].

MIP, MEP, FEC and HGS measurements were performed at all three timepoints. Standard operating procedures were in place to limit inter-assessor variability. All measurements were conducted by the first author or a trained research assistant.

Nutritional status was assessed at T0, with the Short Nutritional Assessment Questionnaire (SNAQ65+). The SNAQ65+ screens nutritional status based on pre-set criteria (involuntary weight loss, upper arm circumference, appetite, and physical function) and distinguishes three categories: undernutrition ('red'), risk of undernutrition ('orange') and no undernutrition ('green') and has shown to have good validity and consistency with mortality in adults [33].

## Study size

No formal sample size calculations were performed a priori for this observational study. As this observational study was conducted parallel to a pilot feasibility study, sample size was determined by—but not limited to—recruitment potential for that study [26].

## Statistical methods

Sample characteristics were analyzed descriptively and expressed as medians (interquartile ranges, IQR). Linear mixed model (LMM) analyses were performed to analyze longitudinal changes in MIP, MEP, FEC and HGS, expressed as regression coefficients with 95% confidence intervals (95% CI) between timepoints and difference in regression coefficients (Δ) between T0-T1 and T1-T2. Longitudinal associations between MIP, MEP, FEC and HGS and

pre-identified predictor variables were investigated in univariate and multivariate mixed model analyses, expressed as regression coefficients (95% CI). Collinearity was assessed by investigating coefficient changes between the univariate and multivariate models. Associations between primary and secondary outcomes were investigated in crude and adjusted models (for age and time dependence) and stratified by sex. Significance levels were set as $p \leq 0.05$.

We performed sensitivity analyses comparing the following data sets: 1) complete versus incomplete cases and 2) participants receiving rehabilitation interventions targeting PICS (REACH program [26]) versus participants receiving usual care. First participants with missing data on primary and/or secondary outcomes at any of the timepoints were analyzed for sample characteristics, predictor variables and outcome data. Independent t-tests or Mann-Whitney U tests (as appropriate) were applied comparing outcome data and covariates of complete and incomplete cases [34] followed by plotting of longitudinal changes in primary and secondary outcomes for complete cases versus data for the total study sample. Similar steps were followed comparing participants receiving the REACH program versus participants receiving usual care with regards to sample characteristics, predictor variables and outcome data. Differences in regression coefficients for primary and secondary outcomes between groups were analyzed using LMM.

Analyses were conducted in IBM® Statistical Package for the Social Sciences (SPSS®), version 27, 2020 for Mac.

### Ethical considerations

Ethical approval was obtained from the Medical Ethics committee at the Amsterdam University Medical Centers (AMC) (2019_012, ABR NL 68475.018.19). Written, informed consent was obtained from all participants.

Guidelines from the STROBE statement were applied for reporting of this study [35].

## Results

Seventy-four potential participants were screened for eligibility. A total of 59 participants (male: 64%, median age [IQR]: 62 [53 to 66]) were included in this study. Seventy percent (n = 41) of the participants were acutely admitted to the ICU and 66% (n = 39) had a cardiorespiratory admission diagnosis. The study sample had a median (IQR) ICU and hospital LOS of 11 (8 to 21) and 35 (21 to 52) days respectively, and median (IQR) duration of MV of 10 (4 to 18) days. In total, forty-three participants (73%) were discharged home. Nineteen participants (32%) received the REACH rehabilitation program (described elsewhere [26]) while 40 participants (68%) received usual care, consisting of rehabilitation interventions in primary care (n = 24) or at a rehabilitation facility (n = 16). At T0, 83% of the sample was classified as having undernutrition (category 'red', SNAQ65+) (Table 1).

### Study flow, drop-out and follow-up

Five participants (9%) withdrew consent during the study; 12% (n = 7) dropped out due to sudden physical deterioration and/or hospital admission and 1 participant could not be contacted for follow-up (Fig 1). Outcome data was obtained from 80% (n = 47) and 75% (n = 44) of the participants at 3- and 6-month follow-up respectively.

### Longitudinal changes in MIP and MEP

For MIP, significant longitudinal changes were observed between each timepoint, while for MEP no significant change was observed after T1 (Fig 2A and 2B). Changes in MIP (95% CI)

**Table 1. Sample characteristics (N = 59).**

| Variable | Outcome |
|---|---|
| **Age** (median/IQR 25–75) | 62 (53–66) |
| **Gender** (n, %) | |
| • Male | 38 (64) |
| • Female | 21 (36) |
| **ICU LOS** (median/IQR 25–75) | 11 (8–21) |
| **Hospital LOS** (median/IQR 25–75) | 35 (21–52) |
| **MV days** (median/IQR 25–75) | 10 (4–18) |
| **Admission category** (n, %) | |
| • Acute | 41 (70) |
| • Elective | 18 (30) |
| **Admission diagnosis** (n, %) | |
| • Respiratory | 26 (44) |
| • Cardiac | 13 (22) |
| • Sepsis | 9 (15) |
| • Oncologic surgery | 10 (17) |
| • Trauma | 1 (2) |
| **Follow-up intervention** (n, %) | |
| • REACH (targeted follow-up) | 19 (32) |
| • Usual care | |
| ○ home discharge | 24 (41) |
| ○ in-patient rehabilitation | 16 (27) |
| **Educational level** (n, %) | |
| • Primary education | 3 (5) |
| • Lower secondary education | 16 (27) |
| • Higher secondary / vocational education | 17 (29) |
| • Higher education | 23 (39) |
| **SNAQ65+ score** (n, %) | |
| • Red (undernutrition) | 49 (83) |
| • Orange (risk of undernutrition) | 6 (10) |
| • Green (no undernutrition | 4 (7) |

IQR: Interquartile Range, LOS: Length of Stay, MV: Mechanical Ventilation, Admission category: relates to an acute or elective ICU admission, Follow-up intervention: REACH = patients discharged home with targeted physical therapy intervention for patients with PICS [26], usual care = home discharge without targeted physical therapy intervention or in-patient rehabilitation

were observed with baseline values being 68.4% (61.2 to 75.7) of predicted, to 91.4% (84.2 to 98.6) at T1 and 98.7% (91.4 to 106.0) at T2. Changes in MEP (95% CI) were observed starting at 76.0% (68.5 to 83.5) of predicted at T0, increasing to 100.9% (93.4 to 108.4) at T1 and 105.5% (97.9 to 113.0) at T2 (Table 2).

## Longitudinal changes in FEC and HGS

Significant longitudinal changes were found for FEC and HGS (Fig 2C and 2D). For FEC, the mean (95% CI) steps improved from 55 (47 to 63) at T0, to 80 (73 to 88) at T1 and 87 (79 to 95) at T2. For HGS, the mean (95% CI) percentage of predicted values changed from 73.3% (66.4 to 80.2) at T0, to 93.9% (87.0 to 100.8) at T1 and 104.7% (97.7 to 111.6) at T2. (Table 2).

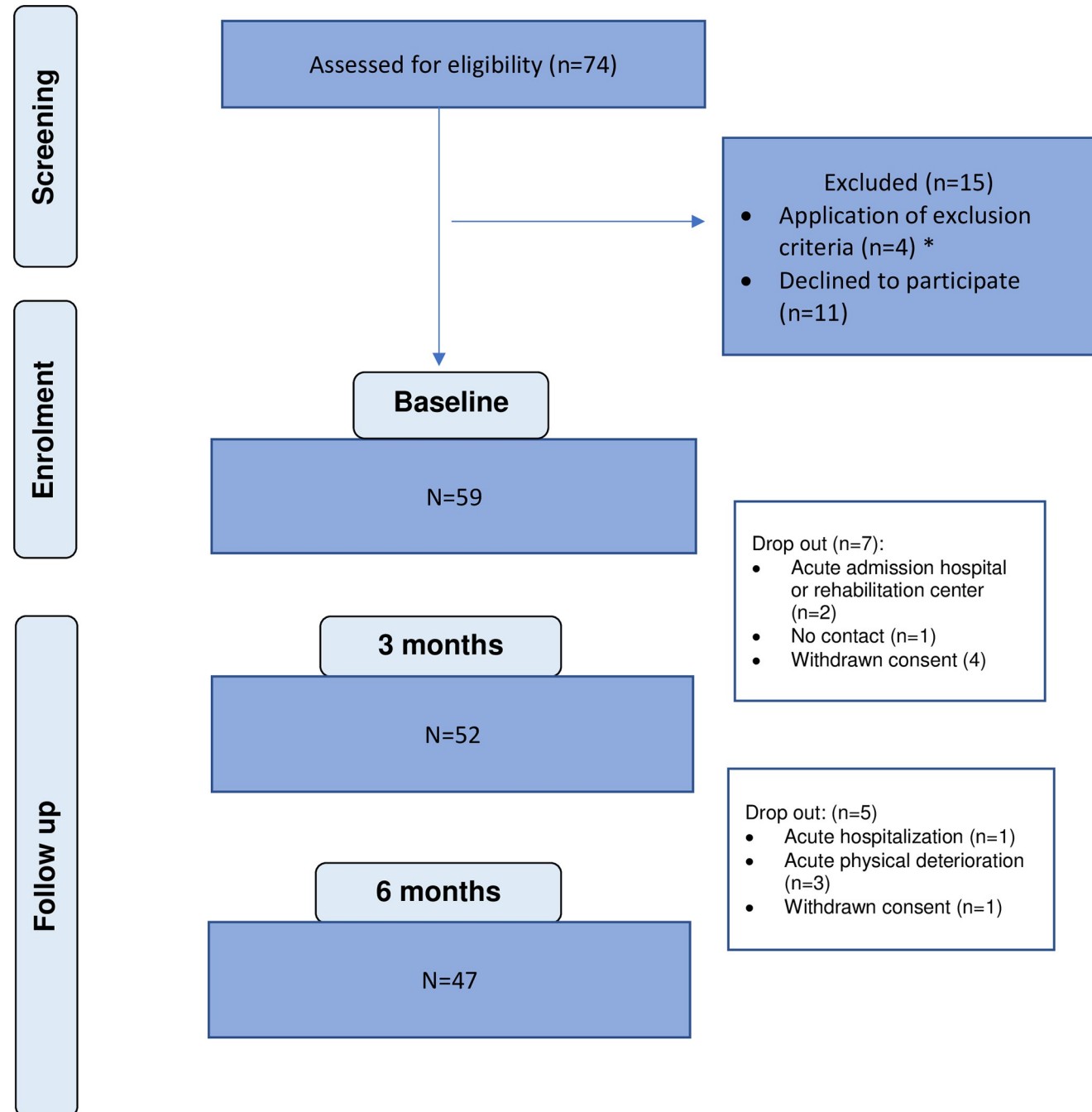

**Fig 1. Consort flow diagram.** *non-invasive ventilation (n = 1), cognitive impairment (n = 1), language barrier (n = 2).

## Associations between predictor variables and primary and secondary outcomes

In both univariate and multivariate analysis, age was identified as a predictor for decreased MIP (% predicted) and FEC, as significant longitudinal associations were found (MIP: -0.6 [-1.2 to -0.1], *p* 0.02 and FEC: -1.03 [-1.5 to -0.5], *p* < 0.001). Hospital LOS was predictive for decreased HGS (% predicted) (-0.5 [-0.7 to -0.2], *p* < 0.001) in both univariate and

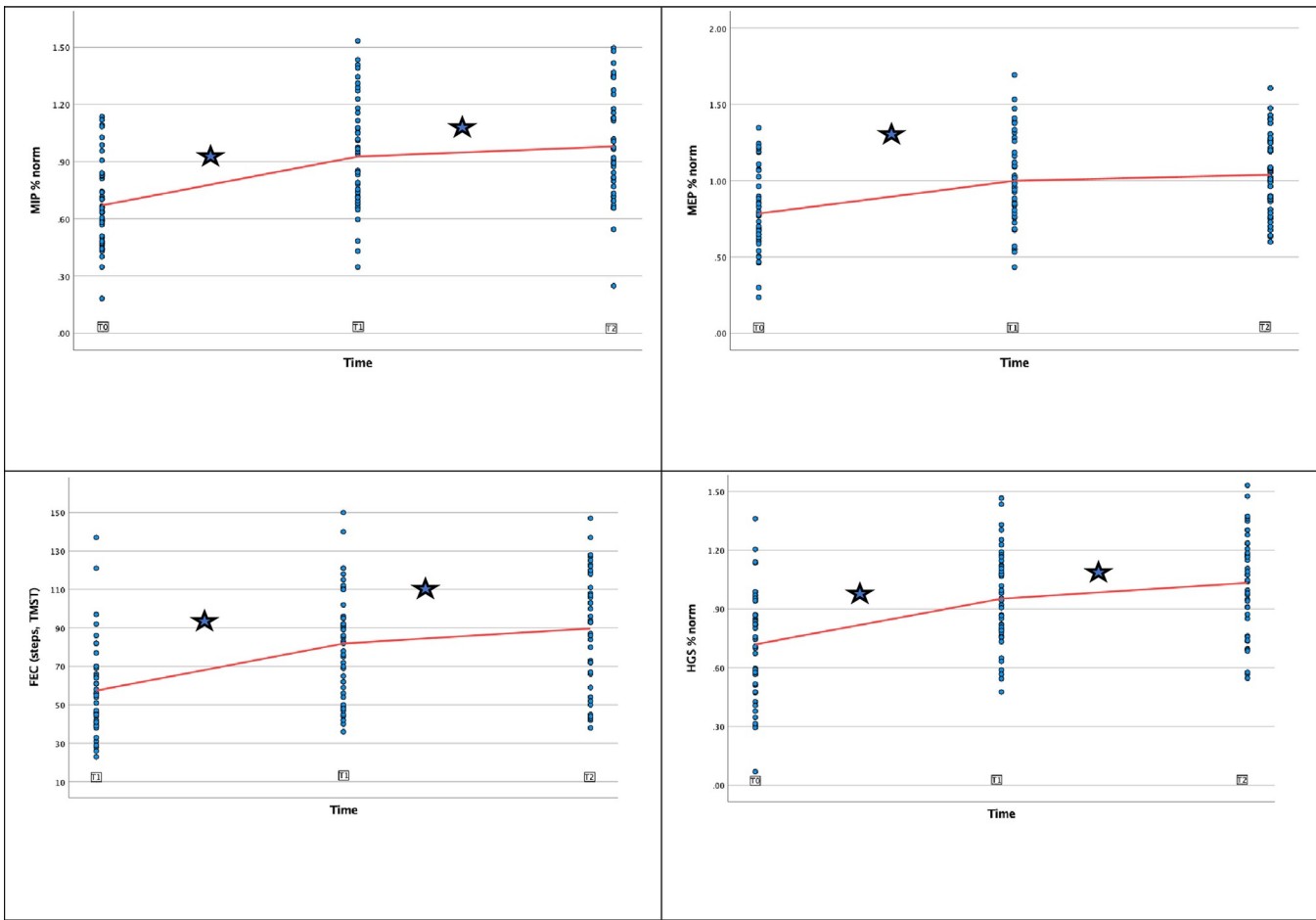

**Fig 2.** Longitudinal changes in functional outcomes: 2a. MIP, 2b. MEP, 2c. FEC, 2d. HGS. Regression coefficients (red line) plotted against observed data (blue dots) at T0, T1 and T2. ★ indicates significant change in regression coefficients between timepoints. Abbreviations: MIP: maximum inspiratory pressure, MEP: maximum expiratory pressure, HGS: Handgrip Strength, FEC: Functional Exercise Capacity, TMST: Two-minute Step Test.

multivariate analysis, while for FEC this longitudinal association was only seen in the univariate model (-0.3 [-0.6 to -0.01] *p* 0.04). Significant longitudinal associations were observed between ICU LOS and HGS (univariate only) and MV days and HGS (both univariate and multivariate) (-0.6 [-1.1 to -0.2], *p* 0.003 and -0.8 [-1.3 to -0.3], *p* 0.001 respectively) (Table 3).

## Longitudinal associations between respiratory muscle weakness and functional outcomes

In both univariate and multivariate models, the observed MIP and MEP (PImax and PEmax, unconverted values) were significantly, longitudinally associated with FEC. As further analyses showed an interaction effect for the sex variable on the (strength of) the association, stratification by sex was applied in the models adjusted for age and time. While for the total sample the longitudinal association between PEmax and FEC remained significant in the age and time adjusted models, it did not for PImax. For PImax and PEmax, significant, longitudinal associations were found with HGS in the model adjusted for time dependency, which did not remain in the age adjusted model. Stratified data shows a difference in the (strength of the) associations for male versus female participants in the models adjusted for age and time, indicative of

**Table 2. Linear mixed model analysis for longitudinal changes in MIP, MEP, FEC and HGS.**

| Variable | T0 | T1 | Δ T1-T0 (95% C.I) | T2 | Δ T2-T1 (95% CI) |
|---|---|---|---|---|---|
| **MIP % predicted cmH2O** | | | | | |
| **β (95% CI)** | 68.4 (61.2 to 75.7) | 91.4 (84.2 to 98.6) | 30.3 (25.2 to 35.3)* | 98.7 (91.4 to 1.06) | 7.3 (2.5 to 12.0)* |
| *p* value | | | < 0.001 | | 0.003 |
| **MEP % predicted cmH2O** | | | | | |
| **β (95% CI)** | 76.0 (68.5 to 83.5) | 100.9 (93.4 to 108.4) | 29.5 (23.6 to 35.3)* | 105.5 (97.9 to 113.0) | 4.6 (-0.9 to 10.1) |
| *p* value | | | 0.001 | | 0.103 |
| **FEC** | | | | | |
| **β (95% CI)** | 54.8 (47.1 to 62.5) | 80.0 (72.5 to 87.5) | 32.2 (25.5 to 38.9)* | 87.0 (79.2 to 94.7) | 7.0 (0.7 to 13.2)* |
| *p* value | | | 0.001* | | 0.029 |
| **HGS % predicted kg** | | | | | |
| **β (95% CI)** | 73.3 (66.4 to 80.2) | 93.9 (87.0 to 100.8) | 31.4 (26.7 to 36.0)* | 104.7 (97.7 to 111.6) | 10.8 (6.3 to 15.2)* |
| *p* value | | | 0.001 | | 0.001* |

β beta regression coefficient Δ change in regression coefficients between timepoints, *significant at **α** 0.05

T0: 1 week after hospital discharge, T1: 3 months after hospital discharge, T2: 6 months after hospital discharge

Abbreviations: CI: confidence interval, MIP: maximum static inspiratory pressure, MEP: maximum static expiratory pressure, FEC: Functional Exercise Capacity, HGS: Handgrip Strength. MIP/MEP and HGS are presented as a percentage of predicted values, FEC expressed in total steps per 2 minutes (Two-minute step test)

an interaction effect of the sex variable on the association. Table 4 provides regression coefficients for the crude and adjusted models.

## Sensitivity analysis results

Sensitivity analysis showed that out of the potential 177 visits, 3 data collection visits were completed according to protocol in 51% (n = 30) of the study sample, 2 out of 3 visits were completed in 32% (n = 19) and in 17.0% (n = 10) only one data collection moment could be established. When analyzing for missing data on primary and secondary outcomes, 12 missing data points (primary outcome: n = 1, secondary outcome [FEC]: n = 11) could be contributed to the fact that safety criteria for testing were not met. Nineteen measurements could not be completed due to restrictions imposed by the national lockdown resulting from the COVID-19 pandemic. When analyzing baseline data for cases with complete follow-up (n = 30) versus

**Table 3. Longitudinal associations between predictor variables and outcomes.**

| Predictor | MIP (% predicted) | MEP (% predicted) | HGS (% predicted) | FEC (steps, TMST) |
|---|---|---|---|---|
| **Hospital LOS** | | | | |
| β (95% CI) | -0.3 (-0.6 to 0.0)† | -0.1 (-0.4 to 0.1) | -0.5 (-0.7 to -0.2)* | -0.3 (-0.6 to -0.01)*‡ |
| **ICU LOS** | | | | |
| β (95% CI) | -0.1 (-0.6 to 0.3) | 0.1 (-0.4 to 0.5) | -0.64 (-1.1 to -0.2)*‡ | -0.4 (-0.9 to 0.08) |
| **MV days** | | | | |
| β (95% CI) *p* value | -0.1 (-0.7 to 0.5) | 0.21 (-0.3 to 0.8) | -0.83 (-1.3 to -0.3)* | -0.33 (-0.9 to 0.2) |
| **Age** | | | | |
| β (95% CI) *p* value | -0.6 (-1.2 to -0.1)* | -0.3 (-0.8 to 0.2) | 0.11 (-0.6 to 0.4) | -1.0 (-1.5 to -0.5)* |

β regression coefficients presenting the univariate association between predictor and outcome variables. CI: Confidence Interval * Significant at **α** 0.05 ‡ Did not remain significant in multivariate analysis † Reached significance in multivariate analysis. Abbreviations: MIP: maximum inspiratory mouth pressure, MEP: maximum expiratory mouth pressure, HGS: Handgrip Strength, FEC: Functional Exercise Capacity, TMST: Two-minute Step Test, LOS: Length of Stay, ICU: Intensive Care Unit, MV: Mechanical Ventilation.

**Table 4. Longitudinal associations between respiratory muscle weakness and functional outcomes, stratified by sex.**

| | Crude | Adjusted for age | Adjusted for time dependency |
|---|---|---|---|
| **PImax∧FEC** | | | |
| β total (95% CI) | 0.69 (0.54–0.84)* | -0.44 (-0.95–0.07) | 6.14 (-4.1–16.3) |
| β ♂ (n = 38) | 0.70 (0.53–0.86)* | -0.73 (-1.31 to -0.16)* | 5.32 (1.02–9.61)* |
| β ♀ (n = 21) | 0.87 (0.54–1.20)* | 0.69 (-0.27 to 1.65) | 3.18 (-0.04 to 6.36) |
| **PEmax∧FEC** | | | |
| β total (95% CI) | 0.52 (0.40–0.64)* | -0.54 (-1.07 to -0.1)* | 18.1 (7.6–28.5)* |
| β ♂ (n = 38) | 0.57 (0.43–0.70)* | -0.77 (-1.33 to -0.20)* | 20.67 (5.22–36.13)* |
| β ♀ (n = 21) | 0.57 (0.29–0.85)* | 0.28 (-0.78 to 1.33) | -3.94 (-21.62 to 13.73) |
| **PImax∧HGS** | | | |
| β total (95% CI) | 0.27 (0.22–0.33)* | -0.08 (-0.30–0.14) | 3.72 (0.87–6.57)* |
| β ♂ (n = 38) | 0.26 (0.19–0.33)* | -0.19 (-0.47 to 0.10) | 5.32 (1.02–9.61)* |
| β ♀ (n = 21) | 0.23 (0.14–0.31)* | 0.06 (-0.19 to 0.31) | 3.18 (-0.004 to 6.36) |
| **PEmax∧HGS** | | | |
| β total (95% CI) | 0.21 (0.17–0.25)* | -0.12 (-0.35 to 0.11) | 4.39 (1.40–7.38)* |
| β ♂ (n = 38) | 0.21 (0.16–0.26)* | -0.20 (-0.51 to 0.11) | 7.68 (2.41–12.94)* |
| β ♀ (n = 21) | 0.16 (0.09–0.23)* | -0.05 (-0.29 to 0.19) | 2.91 (-0.60 to 6.41) |

Independent: PImax and PEmax, uncorrected values, ∧: and β: regression coefficient, ♂: males ♀: females * Significant at **α** 0.05 CI: Confidence Interval. Abbreviations: PImax: unconverted maximum inspiratory mouth pressure, PEmax: unconverted maximum expiratory mouth pressure, FEC: Functional exercise capacity, HGS: Handgrip Strength

cases with incomplete follow-up, participants in the complete group were significantly younger (median age [IQR] 58 [50 to 64] versus 64 [58 to 69], *p* 0.03) but no significant differences were observed for other sample characteristics or outcomes. Comparisons between participants who had received the REACH program (targeting PICS) versus participants receiving usual care (generalized exercise program in primary care or in-patient rehabilitation program) showed no significant differences in longitudinal course (expressed in regression coefficients) for any of the outcomes at any of the timepoints. For sample characteristics, it was observed that participants in the REACH group had a significantly shorter hospital LOS compared to the usual care group (median [IQR] 23 days [14 to 35] versus 42 days [25 to 66]) (S1–S3 Tables, S1 and S2 Figs).

## Discussion

Significant recovery of respiratory muscle strength as well as functional exercise capacity and handgrip strength in the 6 months following ICU and hospital discharge, was observed in our study. At hospital discharge, sample outcomes for respiratory muscle strength were well below predicted values. As MEP recovered to predicted values at 3 months follow up, MIP values remained marginally below predicted at 3- and 6-month follow-up. These results could be indicative of the presence of diaphragm weakness after ICU and hospital discharge [11]. Although the influence of persistent diaphragm weakness on physical recovery after critical illness is not yet fully understood [19], potential contributions to decreased exercise capacity and generalized fatigue can be anticipated. With MIP/MEP and functional outcomes at hospital discharge being markedly lower than population reference values [28, 32, 36] our study sample shows similarities to samples in other observational studies as it included severely deconditioned patients, with potential ICU-AW and possible concurrent ICU-DD [12, 17, 18].

As can be seen from Fig 2A–2D, we observed significant improvements–approaching population reference values—in primary and secondary outcomes over a period of 6 months,

which can possibly be explained by the fact that the larger part of our study sample received some form of rehabilitation intervention after hospital discharge, the fact that our study sample was relatively young, and 68% of the sample completed higher education. Additionally, our study found that older age, longer ICU- and hospital LOS were independent predictors of impeded functional recovery. These findings are supported by several recent publications [37–40]. Our findings are in contrast with other studies investigating functional recovery, such as the cohort of Herridge et al [1]. and Borges et al [41]. When compared to our study results, the cohort in the landmark paper of Herridge et al. [1] comprised a younger group (median age 45), with a median mechanical ventilation duration of 25 days, a longer ICU- and hospital LOS. While the study population showed decreased exercise capacity (compared to predicted) and persistent perceived weakness, there was also a high prevalence of psychological problems (51%), which could possibly be explanatory for worse outcomes at 5 years. Borges et al. [41] reported on a 3-month follow up study in a sample comparable to ours with regards to ICU LOS, but different in that participants were slightly younger and had a shorter duration of MV. Interestingly, this study reports MIP outcomes at hospital discharge and at 3 months, showing results approximating ours at hospital discharge (64% of predicted), but markedly lower outcomes at 3 months (72% of predicted). The difference in observed recovery of MIP over time compared to our study is likely explained by the variety in physical activity levels: while the larger part of our sample received follow-up physical therapy interventions—either in primary care [26] or in a rehabilitation center–the study sample reported by Borges et al. [41] was largely inactive after hospital discharge.

In survivors of critical illness, RMW is sometimes defined as a MIP of $\leq 70\%$ of predicted values [42, 43], but this is not well founded in literature. To determine if RMW was prevalent in our study sample at 1 week after hospital discharge and recovered during the follow-up period, a discussion of the choice of reference values is warranted. As presented by Rodrigues et al. [44] several cut off points for the likelihood of RMW can be identified, varying from 62–77 cmH2O in the younger population ($< 40$ years) to 50–65 cmH2O in the population $> 80$ years old. The European Respiratory Society (ERS) statement on respiratory muscle testing recommends MIP and MEP values to be interpreted in the context of the overall clinical presentation [13], which is more suitable for the post-ICU population [21]. As per these recommendations, we investigated RMW in the context of other physical outcomes which are often impaired after critical illness: exercise capacity and handgrip strength. We found significant associations between parameters of respiratory muscle strength and functional exercise capacity and handgrip strength, associations which remained significant over time. The association between RMW and FEC can be explained by the fact that a functional respiratory muscle pump is required for optimal performance during aerobic exercise [15]. HGS is commonly used as a marker for generalized muscle weakness in survivors of critical illness [31], but this study is the first to show longitudinal associations between RMW and HGS. This potentially justifies continued assessment of MIP and MEP in patients with generalized weakness and deconditioning after ICU and hospital discharge [19, 23]. The associations we found were impacted by sex, in our adjusted models, which could be indicative of different recovery trajectories between male and female patients after critical illness and ICU admission [38].

In patients presenting with a decreased MIP and MEP persisting after ICU discharge, the potential benefit of adding inspiratory muscle training (IMT) to rehabilitation interventions, could be investigated. While recommendations exist for IMT in the critical care setting [20, 21, 45], few studies report on IMT continuing after transfer from the ICU to the hospital ward or beyond. A recent study investigated the effect of IMT up to 2 weeks after extubation in patients who were mechanically ventilated after COVID-19 infection, and found positive effects on dyspnea, pulmonary function, quality of life, and performance on the 6-minute walk test,

when compared to matched controls [46]. Similar effects on exercise capacity have been reported in other populations [47–49] as well as in populations recovering from critical illness [21]. Only 3 studies included in the systematic review by Vorona et al. [21] investigated the potential benefits of IMT after extubation and while MIP improved significantly, the impact of IMT on other clinical outcomes in the post-ICU population remains unknown. As evidence for IMT as treatment modality within post-ICU rehabilitation programs is currently lacking, and considering the findings in our study, we propose to draw from the extensive rehabilitation research in cardiorespiratory patients [50, 51] and to further explore the potential benefits of combined exercise programs targeting respiratory and overall muscle strength as well exercise capacity, in the early recovery phase after critical illness.

## Limitations to our study

**Several limitations to our study can be identified.** First, our recruitment procedure and eligibility criteria might infringe on generalizability of our study results. Recruitment procedures yielded participants who were either transferred to rehabilitation facilities or discharged home with a physical therapy (PT) referral. While it was beyond the scope of this study to determine the contribution of received PT interventions on the recovery in this study's sample, we performed sensitivity analyses and found no differences in outcomes when comparing participants by type or location of received interventions.

Next, due to the design of our study, we did not have access to important patient characteristics such as severity of illness scores, the presence of comorbidities, level of functioning prior to ICU admission or functioning at ICU discharge. This, in combination with the relatively small study sample, should be considered when interpreting our study results.

As is common with prospective cohort studies on vulnerable populations, we performed our analyses on incomplete datasets. Data were missing for several reasons, with the restrictive circumstances during the initial national lockdown in 2020 being the largest contributor. While sensitivity analyses showed no significant differences in outcomes between participants with complete datasets versus the participants with missing data, our results should be interpreted within this context.

No data were available on MIP/MEP prior to or during ICU admission, limiting the interpretation of our findings considering pre-existing functional problems, comorbidities, or severity of the critical illness. Next, we did not explore if MIP was independently associated with MEP, as one recent publication suggests [15]. Considering the pathophysiology of the respiratory muscle pump, it would have been interesting to explore such relations and possible expiratory compensatory mechanisms in the presence of inspiratory muscle weakness. However, our dataset limited us to conduct such analyses.

Data on FEC are expressed in observed values (step count), not corrected for age and gender, as for the population < 60 years, no reference values exist [30, 36]. Step count performance over time is likely confounded by age and/or sex. Unfortunately, data could not be obtained for severely fragile participants. While the test showed responsiveness in detecting change in FEC, especially in the first three months after hospital discharge, a tendency towards a ceiling effect was observed in the second three months. Considering this, we recommend exploring different (functional) exercise capacity tests for patients recovering from critical illness.

## Conclusions

To our knowledge, this is the first study presenting longitudinal data up to 6 months after hospital discharge on the course of recovery of MIP and MEP in patients who received mechanical

ventilation in the intensive care unit. Respiratory muscle weakness was present directly after hospital discharge and MIP remained marginally impaired at 3 months, approaching reference values at 6 months after hospital discharge. As RMW was associated with decreased exercise capacity and handgrip strength, we recommend ongoing assessment of MIP/MEP in deconditioned and weakened patients who are discharged from ICU and hospital and in need of follow-up interventions. More studies are needed to investigate pathophysiological mechanisms explaining associations between RMW, ICU-AW and decreased exercise capacity. For severely deconditioned patients, potential benefits of the addition (or continuation) of inspiratory muscle training as a component of post-ICU exercise programs should be investigated further.

## Supporting information

**S1 Table. Sensitivity analysis: Complete follow-up versus incomplete follow-up.** SD: Standard Deviation, LOS: Length Of Stay, IQR: Interquartile Range, MV: Mechanical Ventilation, SNAQ65+: Short Nutritional Assessment Questionnaire 65+, PImax: maximum static inspiratory mouth pressure, PEmax: maximum static expiratory mouth pressure, GS: Grip Strength, TMST: Two-Minute Step Test. * independent samples t-test ‡ Mann Whitney U test.
(PDF)

**S2 Table. Sensitivity analysis: REACH versus usual care.** SD: Standard Deviation, LOS: Length Of Stay, IQR: Interquartile Range, MV: Mechanical Ventilation, SNAQ65+: Short Nutritional Assessment Questionnaire 65+, PImax: maximum static inspiratory mouth pressure, PEmax: maximum static expiratory mouth pressure, GS: Grip Strength, TMST: Two-Minute Step Test. * Significant between group difference, as tested with the independent samples Mann Whitney U test.
(PDF)

**S3 Table. Sensitivity analysis: Difference in regression coefficients REACH versus usual care.** Results of linear mixed model analysis (LMM) p > 0.05 between REACH versus usual care at any timepoint for all variables. β: beta regression coefficient, CI: Confidence Interval, MIP: maximum static inspiratory pressure, MEP: maximum static expiratory pressure, FEC: Functional exercise capacity, expressed in total steps per 2 minutes (two-minute step test), HGS: handgrip Strength, TMST: Two-Minute Step Test, Kg: kilogram.
(PDF)

**S1 Fig. Sensitivity analysis: Course of recovery total sample versus complete cases (primary outcomes).** MIP: Maximum inspiratory pressure, MEP: Maximum expiratory pressure.
(PDF)

**S2 Fig. Sensitivity analysis: Course of recovery total sample versus complete cases (secondary outcomes).** HGS: Handgrip strength, TMST: Two-minute step test.
(PDF)

**S3 Fig. Sensitivity analysis: Course of recovery REACH versus usual care (all outcomes).** MIP: Maximum inspiratory pressure, MEP: Maximum expiratory pressure, FEC: Functional exercise capacity, TMST: Two-minute step test, HGS: Handgrip strength, Kg: kilogram. T0: baseline, T1: 3 months follow up, T2: 6 months follow up.
(PDF)

## Author Contributions

**Conceptualization:** Mel Major, Stephan Ramaekers, Raoul Engelbert, Marike van der Schaaf.

**Data curation:** Mel Major.

**Formal analysis:** Mel Major, Maarten van Egmond, Marike van der Schaaf.

**Funding acquisition:** Mel Major, Marike van der Schaaf.

**Investigation:** Mel Major.

**Methodology:** Mel Major, Maarten van Egmond, Daniela Dettling-Ihnenfeldt, Stephan Ramaekers, Raoul Engelbert, Marike van der Schaaf.

**Project administration:** Mel Major, Daniela Dettling-Ihnenfeldt.

**Supervision:** Stephan Ramaekers, Raoul Engelbert, Marike van der Schaaf.

**Validation:** Mel Major, Maarten van Egmond, Daniela Dettling-Ihnenfeldt, Raoul Engelbert, Marike van der Schaaf.

**Visualization:** Mel Major.

**Writing – original draft:** Mel Major, Maarten van Egmond, Marike van der Schaaf.

**Writing – review & editing:** Mel Major, Maarten van Egmond, Daniela Dettling-Ihnenfeldt, Stephan Ramaekers, Raoul Engelbert, Marike van der Schaaf.

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
