## [Decision Letter · Decision Letter 0]

5 Aug 2022

PONE-D-22-14900

Course of recovery of respiratory muscle strength and its associations with exercise capacity and handgrip strength: a prospective cohort study among survivors of critical illness

PLOS ONE

Dear Dr. Major,

Thank you for submitting your manuscript to PLOS ONE. After careful consideration, we feel that it has merit but does not fully meet PLOS ONE’s publication criteria as it currently stands. Therefore, we invite you to submit a revised version of the manuscript that addresses the points raised during the review process.

Please revise the text following reviewer's comments

A marked-up copy of your manuscript that highlights changes made to the original version. You should upload this as a separate file labeled 'Revised Manuscript with Track Changes'.An unmarked version of your revised paper without tracked changes. You should upload this as a separate file labeled 'Manuscript'.

We look forward to receiving your revised manuscript.

Kind regards,

Francesco Alessandri

Academic Editor

PLOS ONE

Journal Requirements:

Reviewers' comments:

Reviewer's Responses to Questions

**Comments to the Author**

1. Is the manuscript technically sound, and do the data support the conclusions?

Reviewer #1: Yes

Reviewer #2: Partly

2. Has the statistical analysis been performed appropriately and rigorously? 

Reviewer #1: No

Reviewer #2: Yes

3. Have the authors made all data underlying the findings in their manuscript fully available?

Reviewer #1: Yes

Reviewer #2: Yes

4. Is the manuscript presented in an intelligible fashion and written in standard English?

Reviewer #1: Yes

Reviewer #2: Yes

5. Review Comments to the Author

Reviewer #1: The manuscript entitled "Course of recovery of respiratory muscle strength and its associations with exercise capacity and handgrip strength: a prospective cohort study among survivors of critical illness" is very well written and it is well balanced between the different sections; It is a valuable contribution and its topic is of interest to PLOS ONE.

The only weakness of the paper is that the authors have not considered the singular importance of the sex variable in the prediction of muscle strength. The important characteristics demarcated by literature are the muscle strength differences between male and female, involves biological and cultural issues that determine different levels of muscle strength, which can have repercussions on their recovery after a critical illness. It seems evident from the analyzes presented in table 4, that there is an important modification of the effect in the associations when adjusted for the sex variable, which leads us to identify a possible interaction factor between sex and the outcomes. Therefore, we recommend considering stratification by sex or evaluating the existence of interaction factor and assuming it in the analyses.

We also recommend standardizing the presentation of confidence intervals (CI). The use of the expression "to" to separate the values of the intervals, since there are negative values that can be confused with the "-" sign used in the separation of lower and upper limits.

Reviewer #2: Good morning, first of all I wanted to congratulate you on your scientific paper; I found it very interesting as it deals with aspects of a very important topic on which unfortunately too little is still known (the course of recovery of respiratory muscle strength in patients undergoing mechanical ventilation).

I think your paper can make an interesting contribution to the contemporary scientific literature and poses insights for further study.

In light of this, the changes I would recommend are:

1) As you have already exhaustively described in the paper, the study has some limitations among which the main ones are: the relatively small sample size, the lack of important information on patients' characteristics such as the presence of comorbidities, the presence of diseases and conditions already known prior to ICU admission or developed during ICU stay that may themselves result in reduced MIP, MEP, FEC and HGS, the absence of scores indicating the severity of the main disease for whose patient is admitted to ICU, etc...

Among these limitations, it would be good to add that no difference is made between the type of mechanical ventilation performed on patients in the ICU, particularly whether non-invasive ventilation (NIV) or invasive ventilation (IV) was performed.

There is a huge difference between NIV and IV (which involves oro-tracheal intubation or less frequently naso-tracheal intubation or tracheostomy) in terms of the contribution made by the ventilator to the respiratory muscles during the respiratory cycle and consequently also on the extent of respiratory muscle damage related to mechanical ventilation: during NIV the respiratory muscle work is only partially performed by the ventilator (assisted or supported ventilation), whereas during IV the respiratory muscle work can be either totally (controlled ventilation) or partially performed by the ventilator; moreover, as is logical, during IV the patient is generally more sedated than in NIV, in which more cooperation from the patient is required.

2) Better describe the two minute step test (TMST) in the paragraph "measurements" on page 5. In particular, to be more precise and complete, describe how to obtain for each patient "the set criterion height" (height reached by the right knee of the patient so that the step is considered valid) and what are the contraindications to the test, which in some cases did not allow to perform the test.

3) On page 11, line 261, replace "PImax = observed maximum expiratory mouth pressure" with "PEmax = observed maximum expiratory mouth pressure"

4) On page 12, line 281, add "S3 Fig" to "Supporting information: S1-S3 Table, S1 Fig, S2 Fig."

5) Check and resolve the inconsistency between the MIP at T0 in the abstract "68.1" and in Table 2, page 9 "68.4"

6. PLOS authors have the option to publish the peer review history of their article (what does this mean?). If published, this will include your full peer review and any attached files.

Reviewer #1: No

Reviewer #2: No

---

## [Author Response · Author response to Decision Letter 0]

21 Sep 2022

Please refer to the attached Cover letter, entitled 'Response to Reviewers'

---

## [Decision Letter · Decision Letter 1]

23 Mar 2023

Course of recovery of respiratory muscle strength and its associations with exercise capacity and handgrip strength: a prospective cohort study among survivors of critical illness

PONE-D-22-14900R1

Dear Dr. Major,

We’re pleased to inform you that your manuscript has been judged scientifically suitable for publication and will be formally accepted for publication once it meets all outstanding technical requirements.

Kind regards,

Marjan Mansourian

Academic Editor

PLOS ONE

Reviewers' comments:

 In line 274, Table 4, "PImax" has not been replaced with "PEmax".

**Comments to the Author**

1. If the authors have adequately addressed your comments raised in a previous round of review and you feel that this manuscript is now acceptable for publication, you may indicate that here to bypass the “Comments to the Author” section, enter your conflict of interest statement in the “Confidential to Editor” section, and submit your "Accept" recommendation.

Reviewer #2: All comments have been addressed

2. Is the manuscript technically sound, and do the data support the conclusions?

Reviewer #2: Partly

3. Has the statistical analysis been performed appropriately and rigorously? 

Reviewer #2: I Don't Know

4. Have the authors made all data underlying the findings in their manuscript fully available?

Reviewer #2: Yes

5. Is the manuscript presented in an intelligible fashion and written in standard English?

Reviewer #2: Yes

6. Review Comments to the Author

Reviewer #2: Good evening, I think that the corrections and changes you have made to the text have improved its quality. I just wanted to point out that in line 274, Table 4, "PImax" has not been replaced with "PEmax". I again offer my congratulations for your paper.

7. PLOS authors have the option to publish the peer review history of their article (what does this mean?). If published, this will include your full peer review and any attached files.

Reviewer #2: No

---

## [Editor Report · Acceptance letter]

5 Apr 2023

PONE-D-22-14900R1 

Course of recovery of respiratory muscle strength and its associations with exercise capacity and handgrip strength: a prospective cohort study among survivors of critical illness 

Dear Dr. Major:

I'm pleased to inform you that your manuscript has been deemed suitable for publication in PLOS ONE. Congratulations! Your manuscript is now with our production department. 

Kind regards, 

on behalf of

Professor Marjan Mansourian 

Academic Editor

PLOS ONE